# Characterisation of RSV Fusion Proteins from South African Patients with RSV Disease, 2019 to 2020

**DOI:** 10.3390/v14112321

**Published:** 2022-10-22

**Authors:** Prince Mabilo, Hloniphile Mthiyane, Andiswa Simane, Kathleen Subramoney, Florette Kathleen Treurnicht

**Affiliations:** 1School of Pathology, Faculty of Health Sciences, University of the Witwatersrand, Johannesburg 2193, South Africa; 2Department of Virology, National Health Laboratory Service, Charlotte Maxeke Johannesburg Academic Hospital, Johannesburg 2193, South Africa

**Keywords:** RSV, respiratory, F protein, antigenic sites, palivizumab, South Africa

## Abstract

Respiratory syncytial virus (RSV) is classified into RSV-A and RSV-B, which are further classified into genotypes based on variability in the G gene. The fusion (F) protein is highly conserved; however, variability within antigenic sites has been reported. This study aimed to characterise F proteins from RSV strains detected in South Africa from 2019 to 2020. Patients of all ages, from whom respiratory samples were submitted to the National Health Laboratory Service at Charlotte Maxeke Johannesburg Academic Hospital, South Africa during 2019 to 2020, were included. Complete RSV F genes were amplified for next-generation sequencing. MEGA X software was used for phylogenetic analysis. The overall prevalence of RSV was 5.8% (101/1734). Among 101 RSV positive samples only 69.3% (70/101) were available for characterization of the RSV F protein gene. Among cases included for F gene characterisation, viral co-infections were observed in 50% (35/70) and 25.7% (18/70) were admitted to intensive care units (ICU). About 74.2% (23/31) of F gene sequences cluster with other African NA1/ON1 genotypes. At antigenic site I, the V384I mutation was replaced by V384T in South African strains. The S275F mutation was seen in a single South African strain. The N120 N-linked glycosylation site was present in 25.8% (8/31) of RSV-A F proteins described in this study. For the first time, we detected the rare S275F mutation that is associated with palivizumab resistance.

## 1. Introduction

Respiratory syncytial virus (RSV) causes approximately 3.2 million severe acute respiratory tract infection (SARI)-associated hospitalisations, with overall mortality ranging between 55,000 and 200,000 annually [1,2]. Infection with RSV occurs in 60% of all children before they reach 1 year of age and nearly all children are infected by RSV at least once within 2 years after birth [3]. Over 90% of RSV-associated mortalities are exclusively reported in children less than 5 years of age from developing countries [1,2]. Premature infants and those born from HIV-positive mothers are at high risk of RSV infection, leading to high morbidity and mortality rates in these populations [4,5]. There is currently no licensed vaccine for RSV prevention [6]. Palivizumab (Synagis^®^) is the only prophylactic monoclonal antibody (mAb) treatment available for the prevention of RSV disease, but due to high cost its use is restricted to high-risk infants residing in low-, middle- and high-income countries [7,8,9].

The RSV genome encodes eleven proteins, including attachment (G) and fusion (F) glycoproteins which contain neutralising antibody epitopes capable of inducing a neutralising antibody response, and are targeted in RSV vaccine development strategies [1]. RSV is classified into subtypes A and B, which are characterised by different genotypes based on antigenic and genetic variability of the second hypervariable region (HVR2) of the G protein [1]. Among the two glycoproteins, F protein is the most conserved, with about 90% amino acid sequence identical in both RSV subtypes A and B [10]. Although the F protein is generally conserved, variability in some of the F domains has been observed in the signal peptide, transmembrane domain, and antigenic site Ø [10]. The F protein exists in two conformations (pre-F and post-F) and six antigenic sites (Ø, I to V) are defined. Sites Ø and V are found on the pre-F conformation and have significantly better antibody neutralisation potency than sites I–IV found on the post-F conformation [1]. Site II is the target for palivizumab-binding, site Ø is the target of MEDI8897; site IV is the target of MK-165412; and site V is the target of suptavumab [1]. A study conducted in China observed variability within the antigenic sites, with 56 amino acid differences between the RSV-A and RSV-B F protein [11]. A more recent study that included South African RSV strains from 2018 reported a total of 7 and 11 mutations, respectively, in antigenic sites of the RSV-A and RSV-B F protein sequences at frequencies ranging from 3.3% to 6.7% and 3.0% to 97.0% [1]. This highlights that variability among RSV strains is considerable, and that characterisation of local RSV strains’ F proteins may contribute to a more strategic vaccine design or selection strategy for South Africa. The aim of the study was to characterise F proteins from South African RSV strains during 2019 to 2020.

## 2. Materials and Methods

### 2.1. Setting, Study Population and Sample Size

The study was conducted at Charlotte Maxeke Johannesburg Academic Hospital (CMJAH) in the Department of Virology, National Health Laboratory Services (NHLS) and the University of the Witwatersrand. The study population included patients of all ages, from whom respiratory samples were submitted for respiratory viral infection diagnosis from 16 July 2019 to 31 December 2020. For an expected 6% overall annual prevalence for RSV [12], a population size of about 11.5 million served by CMJAH and 95% confidence interval, a minimum sample size of 87 per annum was estimated [13].

### 2.2. Study Samples

Respiratory samples such as nasopharyngeal swabs, nasal swabs, throat swabs and sputum were tested for RSV and other respiratory viruses with the FastTrack Diagnostics (FTD, Junglinster, Luxembourg) 21-plex respiratory pathogen real-time multiplex PCR assay following total nucleic acid extraction from 200 µL of each sample on the automated MagNA Pure 96 extraction system (Roche Diagnostics, Meylan, France) using the MagNA Pure 96 DNA and Viral NA small volume kit [14]. Stored residual respiratory samples that tested positive for RSV were retrieved for genetic characterisation.

### 2.3. RSV F-Protein Gene Amplification

Viral RNA was extracted from 200µL of each respiratory sample using the QIAamp RNA mini kit (Qiagen, Hilden, Germany) and RNA was eluted in 60 µL of elution buffer. Five microliters of viral RNA from each sample were used to synthesise complementary DNA (cDNA) followed by amplification using the SuperScript III One-step RT-PCR System with Platinum *Taq* polymerase (Invitrogen, Carlsbad CA, USA). The following primers were used to amplify the RSV F protein gene, RSV-A Forward: A5648-S: 5′-GGG GCA AAT AAC AAT GGA GTT-3′(Tm 50.45) and reverse: A7418-AN: 5′-CAT TGT AAG AAC ATG ATT AGG TGC T-3 (Tm 52.76), RSV-B forward: B5502-S: 5′-CGA AAA CAC ACC ACT CCA CAC-3′ (Tm 57.06) and reverse: B7402-AN: 5′- GTG GTT TTT TGT CTA TTT GCT G-3′ (Tm 49.25) [15]. The cDNA synthesis was performed at 50 ℃ for 20 min. The initial denaturation was performed at 94 ℃ for 2 min. This was followed by 40 cycles PCR amplification at 94 ℃ for 20 s, 48 ℃ for 30 s, 68 ℃ for 3 min, followed by final extension at 68 ℃ for 4 min for RSV-A. The PCR amplification cycling conditions for RSV-B were similar to those for RSV-A, except for the annealing temperature, which was set at 55 ℃.

Successfully amplified PCR products were visualised using 1% agarose gel electrophoresis (60 min at 120 V and 400 mA). A 1 kb DNA molecular weight marker (Promega, Madison, USA) was used as a size marker to identify the F gene amplicons with the estimated sizes for RSV-A (1.7 kb) and RSV-B (1.9 kb) PCR products.

### 2.4. Next-Generation Sequencing

Library preparation was performed on F gene specific amplicons using the Nextera DNA prep kit (Illumina, San Diego, CA USA) as per manufacturer’s protocol. Libraries were subsequently normalised to 8 pM and sequenced on the MiSeq system using the MiSeq V2 500 cycle sequencing reagents (Illumina, San Diego, CA USA).

### 2.5. Sequence Read Assemblies and Analysis

The open-source Genome Detective Virus Tool v2.27 [16] was used to assemble sequencing reads generated in FASTQ format to contigs to extract the consensus sequence. Assembled sequence contigs were exported in FASTA format and imported into BioEdit v7.2 Sequence Alignment Editor to align the F gene sequences from this study, together with sequences retrieved from GenBank [17] originating from various countries including the United States (USA), Argentina (ARG), the Netherlands (NLD), Germany (DEU), Belgium (BEL), China (CHN), Vietnam (VNM), Philippines (PHL), Kenya (KEN) and South Africa (RSA). South African sequences from four 2021 strains, of which two were not deposited on any database, were also used (available on the Global Initiative on Sharing Avian Influenza Data [GISAID] database; EPI_ISL_14170477 and EPI_ISL_14170847). Reference sequences used for sequence analysis were the RSV A2 Long strain (Accession number M74568.1 and KT992094.1) and VR-26 strain (Accession number AY911262.1).

### 2.6. Phylogenetic and Genetic Diversity Analysis

Multiple sequence alignments were submitted for phylogenetic tree construction using the MEGA X version 10.2 software [18]. The maximum likelihood method was used to construct a phylogenetic tree using the Hasegawa–Kishino–Yano (HKY) model [19]. The reliability of the phylogenetic tree was estimated by using the bootstrap method with 500 replicates [20]; a cut-off of 60% was applied. This analysis involved 100 nucleotide sequence, each sequence with 1716 positions in the final dataset, and all ambiguous positions were removed using complete deletion method. The RSV A2 Long strain was used as root.

### 2.7. Amino Acid Variation, N-Linked Glycosylation and Impact on Immune Epitopes

The BioEdit v7.2 Sequence Alignment editor was also used to translate nucleotide sequences to amino acids for variation analysis using A2 Long strain (Accession number: M74568.1) and sequences from various countries as references. The HIV Databases [21] were used to determine the number and positions of N-linked glycosylation sites occurring at Asn residues. The consensus sequence to predict potential N-glycosylation sites was presented as the sequon NX(T/S)X where N =Asn, S = Ser, T = Thr and X representing any amino acid except Pro. [22]. Nine-mer cytotoxic T-lymphocytes (CTL) epitopes were predicted using NetTepi-1.0 software [23] estimated from a combined score for binding affinity, peptide stability and CTL propensity for the most prevalent HLA-A (HLA-A01:01, HLA-A02:01, HLA-A03:01, HLA-A11:01, HLA-A24:02, HLA-A26:01) and HLA-B (HLA-B07:02, HLA-B15:01, HLA-B27:05, HLA-B35:01, HLA-B39:01, HLA-B40:01 and HLA-B58:01) alleles [24,25]. To obtain a global % rank score peptides were compared against 200 000 natural peptides [24]. Strong binders were identified as peptides with % rank of < 0.8% [26]. Similarly, the A2 Long strain was used as a reference against sequences from our study together with other South African sequences.

### 2.8. Statistical Analyses

Data analysis was performed using Statistica version 14.0.0.18. Descriptive statistics were presented as frequencies and proportions to summarise data distribution for demographic and clinical variables.

### 2.9. Ethics

The study was approved by the Human Research Ethics Committee (HREC) of the University of the Witwatersrand (M201161), Johannesburg, South Africa.

## 3. Results

### 3.1. Demographic Characteristics of Study Population

Between 16 July 2019 to 31 December 2020, 1734 respiratory samples were tested for respiratory viruses, of which 39.4% (684/1734) were from 2019 and 60.6% (1050/1734) from 2020. The overall prevalence of RSV was 5.8% (101/1734), of which 70 (69.3%) RSV positive cases with residual stored samples could be characterized genetically (Table 1). Males accounted for 48.6% (34/70) of cases, females for 42.9% (30/70) and data on gender classification was missing for 8.6% (6/70). Distribution of the patients by age group showed that most patients (75.7%; 53/70) were aged ≤2 years, (4/70; 5.7%) were >2 to 5 years, (2.9%; 2/70) were >5 years to <65 years old (Table 1).

### 3.2. Other Respiratory Virus Infections

Among the 70 RSV-positive cases included for genetic characterisation, co-infection with other respiratory viruses were observed in 35 (50.0%) (Table 1). Most 28.6%; 20/70) were co-infected with one respiratory virus, 14.3% (10/70) were co-infected with two respiratory viruses and 7.1% (5/70) were co-infected with three or more respiratory viruses. Admission to intensive care units (ICU) were reported for 18/70 (25.7%) cases. Among ICU admissions, 38.9% (7/18) of cases were co-infected with another respiratory virus. Rhinovirus was the most frequently detected co-infecting virus (57.1%; 4/7), followed by human adenovirus (42,9%; 3/7), enterovirus (28,6%; 2/7), and bocavirus at (14.3%; 1/7).

### 3.3. Phylogenetic Analysis of RSV F Genes

F gene sequence data were generated for (91.2%; 31/34) samples and only RSV-A strains were identified in 2019–2020. Maximum likelihood tree analysis was conducted and the majority (74.2%; 23/31) of sequences from this study clustered together with bootstrap values ranging from 64% to 99% (Figure 1). Meanwhile, other sequences clustered with other South African sequences from 2018 and 2021 that were NA1-like. It should be noted the genotype assignment was based on G protein gene genotyping, and ON1 genotypes fall within the NA1 group (Figure 1). Strains previously genotyped as GA2 had F genes that formed two strongly supported sub-clusters within the main NA-1 cluster and these included strains from Kenya sampled in 2011 to 2015 (Figure 1). We did not see close clustering of South African RSV F protein genes with sequences from other countries. The RSV F gene sequences were uploaded on the GISAID database and accession numbers for download by registered users are provided (Appendix A).

### 3.4. Amino Acid Variations at Antigenic Sites of the F Protein

Ten amino acid changes were detected in four antigenic sites (I, II, IV and V) when compared with global reference sequences from 2001 to 2021, with frequencies ranging from 0.3 to 100% (Table 2). Five amino acid changes were detected for the majority of sequences at high frequencies at the following positions: I379V (100%; 329/329) and V384I (90.8%; 299/329) at antigenic site I, N276S (20.1%; 66/329) at antigenic site II, M447V (100%; 329/329) at antigenic site IV and V152I (93%; 307/329) and L178V (100%; 329/329) at antigenic site V. V152I amino acid change was prevalent in RSV-A strains from all countries. N276S was prevalent among South African sequences from this study at a frequency of (94.3%; 51/54). Amino acid change V384T (7.3%; 24/329) at antigenic site I and S275F (0.3%; 1/329) at antigenic site II were exclusive to South African sequences. No amino acid changes were detected at site Ø and III (Table 2).

### 3.5. N-Linked Glycosylation and CTL Epitope Predictions among Deduced F Proteins

A total of 6 N-glycosylation sites (N27, N70, N116, N120, N126 and N500) were predicted and similar between our sequences and reference sequences. Five N-glycosylation sites were predicted at high frequency at positions N27, N70, N116, N126 and N500 (Figure 2). Gain of the potential glycosylation site at position 120 was identified in 25.8% (8/31) of 2020 sequences from this study (Figure 2). This gain was also observed in South African reference sequences from 2018 and 2021 at frequencies of 100% (19/19) and 25.0% (1/4), respectively.

The following mutations were observed in the RSV F protein signal peptide sequences: A8T, T16A/I, T12I/A and F20L. The V152I cytotoxic T lymphocyte (CTL) escape mutation in epitope SAIASG**V**AV restricted by HLA-B07:02 was observed in 2018 to 2021 strains from South Africa (Table 3). The Y33H escape mutations were seen in epitopes F**Y**QSTCSA and GQNITEEF**Y** restricted by HLA-A24:01 and HLA-B15:01, respectively. The S275F mutation was located in the HLA-A11:01 restricted epitope M**S**NNVQIVR at antigenic site II. CTL escape mutation K123Q in the MNYTLNNA**K** epitope (restricted by HLA-A03:01; HLA-A11:01) among strains from 2019 and 2020 was associated with a reversion (T > A) mutation at position 122 (Table 3). Among South African strains from 2018, an A23T mutation was seen in the LTAVTFCF**A** epitope; however, this mutation reverted in almost all South African strains from 2019 to 2020.

## 4. Discussion

The F protein of RSV is a key immune response determinant and target for antiviral and vaccine development, due to its conserved nature when compared to the G surface glycoprotein. Several mutations have been observed in the F gene between different RSV geographic strains [11,15]. This study was one of the few studies in South Africa to report on the molecular epidemiology of the RSV F protein. In this study, conducted from 16 July 2019 to 31 December 2020, we mainly identified F gene sequences characterised as belonging to the RSV-A subtype. This may be due to the predominance of RSV-A strains in South Africa during this period. Both RSV-A and RSV-B have been found to co-circulate within seasonal epidemics in South Africa [27]. Respiratory virus co-infections and ICU admissions were common among infants that were predominantly ≤2 years of age, accounting for 33.3% of co-infections. The influence of RSV on ICU admissions is inconclusive, as few studies found that co-infections increase ICU admissions, while others report a decrease in ICU admissions in the presence of co-infections [28,29,30].

Most candidate vaccines for RSV are based on the historical RSV A2 Long strain, which were used in this study as a reference [10]. Phylogenetic analysis showed that F gene sequences from this study are similar to South African strains from 2018 and Kenyan strains that were genotyped mainly as NA1 or ON1. The South African strains from 2018 were also genotyped as ON1 [27]. The characteristic 72-nucleotide duplication in the HVR2 of G protein of ON1 is suggested to increase viral replication, which thus contributed to its current widespread dominance [1].

Overall, the diversity of RSV F protein in South African strains was extremely low, with 5.3% positions with variations. Antigenic sites Ø and V are prone to mutations because they elicit high potency of neutralising antibodies [27]. However, no changes were observed at antigenic site Ø. Antigenic site V had two highly frequent amino acid changes, V152I and L178V. These changes were detected in strains from all featured countries. In contrast, antigenic site I elicited low potency neutralising antibodies and was previously reported as unlikely to be prone to mutations [31]. However, in this study the highest number of mutations were found in antigenic site I. In South African sequences, the wild-type amino acid change V384I in antigenic site I was replaced by the V384T mutation. Amino acid change N276S at antigenic site II has been reported to affect binding. The high frequency of N276S in antigenic site II was similar to what was observed among South African sequences from 2018 and, in addition, we also observed the S275F mutation among South African strains from this study. The N276S mutation at antigenic site II is located in the binding site for palivizumab. This highly prevalent N276S mutation is not associated with palivizumab resistance [15]. However, here we report for the first time the rare S275F mutation which is one of four single amino acid mutations that cause palivizumab resistance in vitro [32]. Therefore, there is a need to monitor the prevalence of these naturally resistant RSV strains through continued genomic surveillance to inform therapy and vaccine strategies.

We confirm that both the position and number of N-glycosylation sites across the F protein are highly conserved, suggestive of the importance of these glycosylation sites on the folding and biological activity of the F protein [33]. However, the importance of N-glycosylation for RSV F protein functionality is still unclear, although it was indicated that it may impact fusion activity [33,34]. The ability of a virus to change N-glycosylation patterns is one of the mechanisms for immune response evasion [5,35]. In this study, we show loss of the NXT site at position N120 in South African RSV-A strains detected after 2018, whereas the same was also reported for RSV strains from China sampled from 2003 to 2014 [36]. This may indicate that antibody-based immune selection pressure may be a key driver of evolution of F in this region, although the role of CTL pressure cannot be overruled.

Several amino acid mutations occurred in the signal peptide region of the F proteins. Mutations in almost all of these signal peptide positions were described previously, however, its impact on F protein expression and functionality were not discussed [10]. The concomitant reversion observed at position 122 with the escape mutation at position 123 may indicate the importance of these positions for maintaining functionality of the fusion peptide. These fusion peptide mutations are also thought to be associated with the helix structure which forms the antibody binding pocket for the pre-F D25 monoclonal antibody [37].

In conclusion, we show that despite the conserved nature of the RSV F protein among South African strains, amino acid changes potentially associated with CTL and antibody escape were identified. This confirms that continued immune adaptation is responsible for the evolution of the RSV F protein genes.

## Figures and Tables

**Figure 1 viruses-14-02321-f001:**
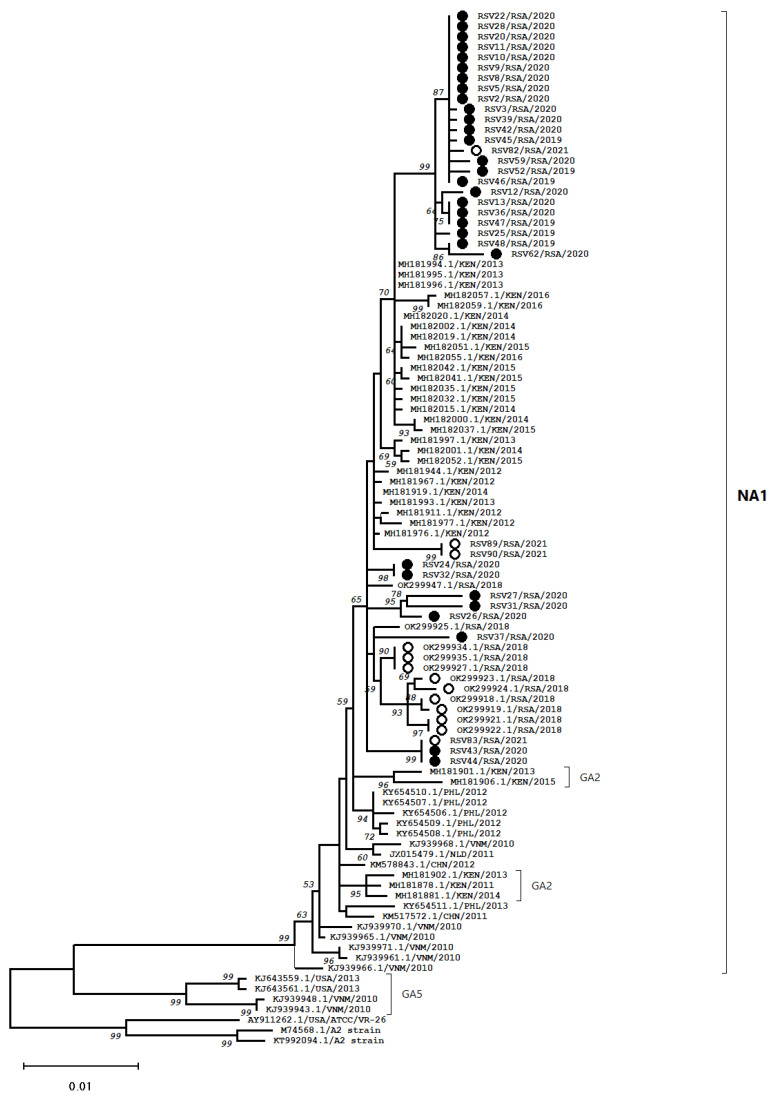
Maximum-likelihood phylogenetic tree based on the 1716 bp long F gene sequences of RSV-A strains from South Africa that circulated in 2019 and 2020 (filled black circles). Reference South African strains are indicated by open black circles. A2 Long strains were used as root. Genotype classification used are based on G protein gene sequence genotypes assigned to reference sequences.

**Figure 2 viruses-14-02321-f002:**
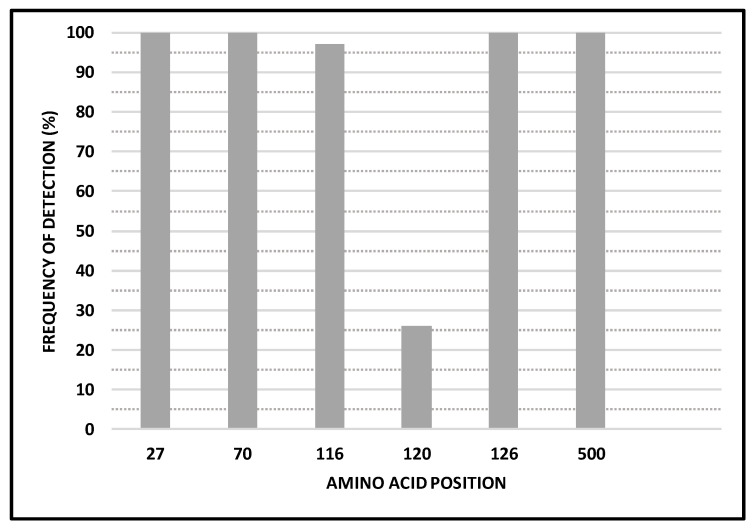
A graphic representation of N-glycosylation site positions across the South African RSV F protein sequences and the frequency of detection of NX(T/S)X sequons, 2019 to 2020.

**Table 1 viruses-14-02321-t001:** Demographic and clinical characteristics of the RSV-positive patients with available respiratory samples following diagnostic testing at the National Health Laboratory Service, Charlotte Maxeke Johannesburg Academic Hospital, Johannesburg, South Africa, 2019 to 2020.

Variable	2019 N (%)	2020 N (%)	Total N (%)
**Age group** ≤2 years >2 to 5 years >5 to <65 years ≥65 years Missing Total	13 (68.4) 0 (0) 2 (10.5) 0 4 (21.1) 19 (100)	40 (78.4) 4 (7.8) 0 (0) 0 (0) 7 (13.7) 51 (100)	53 (75.7) 4 (5.7) 2 (2.9) 0 (0) 11 (15.7) 70 (100)
**Gender** Female Male Missing Total	5 (26.3) 11 (57.9) 3 (15.8) 19 (100)	25 (49.0) 23 (45.1) 3 (5.9) 51 (100)	30 (42.9) 34 (48.6) 6 (8.6) 70 (100)
**Ward** ^a^ICU Non-ICU Missing Total	5 (26.3) 12 (63.2) 2 (10.5) 19 (100)	13 (25.5) 36 (70.6) 2 (3.9) 51 (100)	18 (25.7) 48 (68.6) 4 (5.7) 70 (100)
**Viral co-infections** 0 1 2 ≥3 **Total**	11 (57.9) 5 (26.3) 2 (10.5) 1 (5.3) 19 (100)	24 (47.1) 15 (29.4) 8 (15.7) 4 (7.8) 51 (100)	35 (50) 20 (28.6) 10 (14.3)) 5 (7.1) 70 (100)

^a^ICU: Intensive care unit.

**Table 2 viruses-14-02321-t002:** Frequency of mutations within the RSV F protein antigenic sites in South African and strains from other countries (N = 329).

Antigenic Site	Positions	Mutation	^a^ Frequency of Detection, n (%)	* Countries Where Detected
**I**	27–45; 312–318; 378–389	** *Y33H* **	** *1 (0.3)* **	RSA
		I379V	329 (100)	RSA, USA, NLD, ARG, DEU, CHN, PHL, VNM, BEL, KEN
		V384I	299 (90.8)	RSA, USA, NLD, ARG, DEU, CHN, PHL, VNM, BEL, KEN
		** *V384T* **	** *24 (7.3)* **	RSA
**II**	254–277	** *S275F* **	** *1 (0.3)* **	RSA
		** *N276S* **	** *66 (20.1)* **	RSA, NLD, CHN, PHL, VNM, KEN
**IV**	422–471	M447V	329 (100)	RSA, USA, NLD, ARG, DEU, CHN, PHL, VNM, BEL, KEN
**V**	55–61; 146–194; 287–300	V152I	307 (93)	RSA, USA, NLD, ARG, DEU, CHN, PHL, VNM, BEL, KEN
		L178V	329 (100)	RSA, USA, NLD, ARG, DEU, CHN, PHL, VNM, BEL, KEN

^a^ Amino acid changes detected at low frequency are indicated in bold-italics. * RSA: Republic of South Africa is underlined; USA: United States of America; ARG: Argentina; NLD: The Netherlands; DEU: Germany; PHL: Philippines; VNM: Vietnam; CHN: China; BEL: Belgium; KEN: Kenya.

**Table 3 viruses-14-02321-t003:** CTL escape mutations in predicted RSV F epitopes.

Position	HLA Restriction	* Peptide (Amino Acid Change Position in Bold)	Amino Acid Change and Frequency (%)	Detected in South African Strains
25–33	HLA-B15:01	GQNITEEFY	G25S (100); Y33H (1.85)	2018
32–40	HLA-A24:01	FYQSTCSAV	Y33H (1.85)	2018
114–122	HLA-A02:01	FMNYTLNNA	A122T (51.9)	2018, 2020, 2021
115–123	HLA-A03:01; HLA-A11:01	MNYTLNNAK	K123Q (55.5)	2019, 2020, 2021
133–141	HLA-B27:05	RKRRFLVFL	V139G (100)	2018, 2019, 2020, 2021
146–154	HLA-B07:02	SAIASGVAV	V152I (100)	2018, 2019, 2020, 2021
274–282	HLA-A11:01	MSNNVQIVR	S275F (1.85)	2019

* A2 strain sequence was used as reference.

## Data Availability

Sequence data are available on GISAID (https://gisaid.org/).

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
