# Peer review of "Characterisation of RSV Fusion Proteins from South African Patients with RSV Disease, 2019 to 2020"

_viruses, 2022, doi:10.3390/v14112321_

Round 1
Reviewer 1 Report
The manuscript by Mabilo et al. provides a glimpse of recent antigenic diversity within the fusion protein of RSV strains circulating in Johannesburg, South Africa. Work of this kind is of potential importance to vaccine design and, if repeated using samples from the same location and/or compared to similar results from other countries, can help us understand how RSV is evolving.
Minor issues:
Abstract:
Line 12: change ‘were’ to ‘has been’
Line 15: remove comma after ‘during’
Lines 17-19 are confusing and need to be fixed. 69.3% of RSV-positive samples were sequenced, no? This needs to be indicated
Line 20: change ‘is replaced’ to ‘was replaced’
Line 23: change ‘are’ to ‘is associated’
Introduction:
Line 28: change ‘infections’ to ‘infection’
Line 36: remove comma after ‘(Synagis)’
Line 54: remove comma after ‘China’
Materials and Methods:
Line 95: remove ‘to’ from ‘to 1%’
Results:
Change lines 151- to: ‘…were tested for respiratory viruses, of which 39.4% (684/1734) were from 2019 and the remaining (1050/1734) were from 2020. The overall prevalence of RSV was 5.8% (101/1734), and a total of 69.3% (70/101) of RSV-positive cases had residual stored samples that could be characterized genetically…’
In general, the authors need to make the Demographic characteristics of study population subsection more readable by editing the text in ways consistent with what I’ve provided above.
Question: Was F gene sequence data generated only for RSV-positive samples without co-infection? Why not both?
Please define CTL.
Discussion:
Line 241: remove comma after ‘F gene’
Line 253: change ‘were’ to ‘was’
Line 254: change ‘is’ to ‘are’
Author Response
Reviewer 1
Abstract
- Line 12: “were” replaced with “has been”
- Line 15: A comma was removed after “during”
- Lines 17-19: Sentence was revised to read “Among 101 RSV positive samples only 69.3% (70/101) were available for characterization of the RSV F protein gene. Among cases included for F gene characterisation viral co-infections were observed in 50% (35/70) and 25.7% (18/70) were admitted to intensive care units (ICU).”
- Line 20: replaced “is” with “was”
- Line 23: “are” was replaced with “is”
Introduction
- Line 28: “infections” was replaced with “infection”
- Line 36: Comma after “(Synagis)” was removed
- Line 54: Comma after “China” was removed
Materials and methods
- Line 95: “to” was deleted
Results
- Line 151-155: Section was revised as advised to read “1734 respiratory samples were tested for respiratory viruses, of which 39.4% (684/1734) were from 2019 and 60.6% (1050/1734) from 2020. The overall prevalence of RSV was 5.8% (101/1734), of which 70 (69.3%) RSV positive cases with residual stored samples could be characterized genetically”
- Question on viral co-infections: Table 1 was updated for viral co-infections and related results in text were revised for clarity. Samples with and without other viral co-infections were included for amplicon-based sequencing.
- Line 226: CTL was defined.
Discussion
- Line 241: Comma after “F gene” was deleted
- Line 253: “were” replaced with “was”
- Line 254: “is” replaced with “are”
Reviewer 2 Report
See attached pdf of the manuscript with several comments

Author Response
Reviewer 2
Abstract
- Line 17 -22: Sentences were revised to read: “Among 101 RSV positive samples only 69.3% (70/101) were available for characterization of the RSV F protein gene. Among cases included for F gene characterisation viral co-infections were observed in 50% (35/70) and 25.7% (18/70) were admitted to intensive care units (ICU).”
- Line 22: “About” was inserted before “74.2%”
- Line 23-23: Section was revised to read: “V384I mutation was replaced by V384T in South African strains. … N120 N-linked glycosylation site was present in 25.8% (8/31) of RSV-A F proteins described in this study.
- Line 23: “are” is replaced with “is”
- Line 26: see point 3 above.
Introduction
- Line 30-31: Sentence was revised to read “Infection with RSV occurs in 60% of all children before they reach 1 year of age and nearly all children are infected by RSV at least once within 2 years after birth”
- Line 31-32: Revised to read “Over 90% of RSV associated mortalities are exclusively reported in children less than 5 years of age from developing countries”
- Line 32-33: Sentence on “all cause SARI” was deleted.
- Lines 38-39: Sentence was revised to read “to high cost its use is restricted to high-risk infants residing in low-, middle- and high-income countries [7,8,9].”
- Line 42: References’ numbering and reference list was amended
- Lines 60-61: “The study aimed” was revised to read “The aim of the study was”
Materials and methods
Lines 113-114: Sentence was revised to read “South African sequences from four 2021 strains of which two were not deposited on any database were also used (available on the Global Initiative on Sharing Avian Influenza Data [GISAID] database; EPI_ISL_14170477 and EPI_ISL_14170847).”
Results
- Lines 155-156: Sentence was revised to read “Males accounted for 48.6% (34/70) of cases, females for 42.9% (30/70) and data on gender classification was missing for 8.6% (6/70)”
- Table 1: “Co-infections” changed to “Viral co-infections”. Table was also updated to also include the cases without viral co-infections.
- Lines 165-169: Section was revised to read “Among the 70 RSV positive cases included for genetic characterisation, co-infection with other respiratory viruses were observed in 35 (50.0%) (Table 1). Most 28.6%; 20/70) were co-infected with one respiratory virus, 14.3% (10/70) were co-infected with two respiratory viruses and 7.1% (5/70) were co-infected with three or more respiratory viruses.”
- Lines 168-170: The sentence was revised to read “Admission to intensive care units (ICU) were reported for 18/70 (25.7%) cases. Among ICU admissions, 38.9% (7/18) of cases were co-infected”
- Lines 184-192: Section revised as follow “the GISAID database and accession numbers for download by registered users are provided (Supplementary table S1)..”
- Table S1 was inserted after the references.
- Line 207: Figure 1-quality was improved.
- Figure 2: quality of figure was improved and revised to only display data from this study.
- Line 220-221: Section was revised to read “Gain of the potential glycosylation site at position 120 was identified in 25.8% (8/31) of 2020 sequences from this study (Figure 2). This gain was also observed South African reference sequences from 2018 and 2021 at frequencies of 100% (19/19) and 25.0% (1/4), respectively.”
- Line 223-224: The legend of figure 2 was amended to read “A graphic representation of N-glycosylation site positions across the South African RSV F protein sequences and the frequency of detection of NX(T/S)X sequons, 2019 to 2020”
Discussion
- Line 283: Reference numbering was amended.
References
- Line 331-333: Reference was replaced.
- Line 334-336: Reference was deleted
- Line 337-350: Reference numbering was updated
- Line 351: A reference was added
Supplementary data
- Supplementary table 1 was added after references